# Effects of Physical Activity Interventions in the Elderly with Anxiety, Depression, and Low Social Support: A Clinical Multicentre Randomised Trial

**DOI:** 10.3390/healthcare10112203

**Published:** 2022-11-02

**Authors:** Anna Ruiz-Comellas, Glòria Sauch Valmaña, Queralt Miró Catalina, Isabel Gómez Baena, Jacobo Mendioroz Peña, Pere Roura Poch, Anna Sabata Carrera, Irene Cornet Pujol, Àngels Casaldàliga Solà, Montserrat Fusté Gamisans, Carme Saldaña Vila, Lorena Vázquez Abanades, Josep Vidal-Alaball

**Affiliations:** 1Primary Care Center (PCC) Sant Joan de Vilatorrada, 08250 Sant Joan de Vilatorrada, Spain; 2Health Promotion in Rural Areas Research Group, Institut Català de la Salut, 08272 Sant Fruitós de Bages, Spain; 3Central Catalonia Research Support Unit, Jordi Gol i Gurina University Institute for Research in Primary Health Care Foundation, 08007 Barcelona, Spain; 4Faculty of Medicine, Universidad de Vic–Universidad Central de Catalunya, 08500 Vic, Spain; 5Spain Epidemiological Surveillance and Response to Public Health Emergencies Service, Public Health Agency of Catalonia, 08005 Barcelona, Spain; 6Consorci Sanitari de Vic, Vic Hospital, 08500 Vic, Spain; 7PCCNavàs, 08670 Navàs, Spain; 8PCCSúria, 08260 Súria, Spain

**Keywords:** physical activity, exercise, elderly, anxiety, depression, social support, mental health, primary healthcare

## Abstract

The percentage of older people is increasing worldwide. Loneliness and anxious–depressive states are emerging health conditions in this population group, and these conditions give rise to higher morbidity and mortality. Physical activity (PA) and social relationships have been linked to physical and mental health. The objective of this study was to evaluate whether a 4-month programme of moderate PA in a group would improve the emotional state, levels of social support, and quality of life in a sample of individuals >64 years of age. A multicentre randomised clinical trial was designed in primary care. Ninety (90) participants were selected. After the intervention, there were positive differences between the groups, with significant improvements in the intervention group (IG) in depression, anxiety, health status perception, and social support. Walking in a group two days per week for 4 months reduced clinical depression and anxiety by 59% and 45%, respectively. The level of satisfaction was very high, and adherence was high. In conclusion, the moderate group PA programme improved clinical anxiety, depression, social support, and perceptions of health status in the patients studied.

## 1. Introduction

The percentage of people over 64 years of age in Spain in 2021 was 19.9% (9.38 million people) [1] and is expected to increase further to 25.2% by 2033, according to figures published by the National Institute of Statistics (INE) [2]. Promoting quality of life in this population is now a priority for health researchers.

Health-related quality of life (HRQoL) is defined as the perception of the physical, mental, and social effects of illness on wellbeing [3]. HRQoL is considered to be an important health outcome among the population and an essential public health tool for assessing their physical and social functioning, mental health, and wellbeing, as well as for evaluating population-based intervention programmes [4]. Poor HRQoL perceptions have been associated with older age, lack of social support, high levels of depression, low self-esteem, low social class, sex (female), chronic medical conditions, high body mass index, and sedentary lifestyles [5].

In our society, loneliness and problems of depression and anxiety are common among the elderly. In Western countries, loneliness has a prevalence of 24–40% in people over 65 years of age and increases with age [6,7,8,9]. Research has linked social isolation and loneliness to increased risks of physical and mental illnesses [10]. The prevalence of depression and chronic anxiety in Spain in 2020 stood at 16.26% and 9.08%, respectively, in subjects over 64 years old [11]. There is a growing need to identify effective interventions that address these problems, as there are often limited pharmacological options available for these patients.

Various studies have revealed that physical activity (PA) is one of the most important resources to support health [12] and HRQoL [13,14] in older adults—not only physically, but also emotionally [15], reducing depression [16,17,18,19] and anxiety [20,21]. In addition, the dynamics and social cohesion of walking groups can have an effect on fostering and maintaining adherence and positive attitudes toward physical activity [22], companionship, and the shared experience of wellbeing [17,23].

The high variability of different PA intervention protocols may explain why the results on the relationship between PA and psychological wellbeing and HRQoL are not consistent. Further research is needed to confirm this effect.

The main objective of this study was to evaluate whether a 4-month group programme of moderate physical activity could improve the emotional state, levels of social support, and quality of life of a sample of individuals over 64 years of age with anxiety, depression, or low social support.

## 2. Materials and Methods

The design used was that of a multicentre randomised clinical trial with two groups, with a follow-up period of 1 year.

The study population consisted of patients over 64 years of age assigned to 3 basic health areas: Sant Joan de Vilatorrada (12,721 users), Súria (8956 users), and Manresa (23,351 users). These areas were chosen to study patients living in urban and semi-urban areas.

The inclusion criteria were as follows: (1) having a score ≥14 and <28 on the Beck Depression Inventory (BDI-II), a score ≥10 on the Generalised Anxiety Disorder Scale (GAD-7), or a score ≥32 on the DUKE-UNC-11 Social Support Scale; (2) the possibility of follow-up for one year with the same primary care team; (3) the ability to read and write in Spanish or Catalan; and (4) the possibility of going for a walk for 1 h per day, two days per week.

The exclusion criteria were as follows: (1) a diagnosis of dementia or moderate cognitive impairment; (2) a diagnosis of major depression (Beck Depression Inventory (BDI-II) score ≥ 28); (3) dependency disorders due to chronic pathology or abuse of alcohol or other drugs; (4) receiving psychological therapy from mental health specialists; (5) temporary physical or mental impairment preventing walking for 1 h per day, two days per week; and (6) not signing the informed consent form. 

### 2.1. Intervention

The intervention group participated in a moderate-intensity aerobic PA programme (3 METs [24]), which consisted of walking 2 days per week in a group for 4 months. Two health workers accompanied the group and supervised the participants, advising on the pace of the walk and the distance to be covered, depending on the physical capacity of each participant and their underlying pathologies. The intensity of the PA was controlled by means of the conversation test. As there were two companions, one was at the front with the group that was in better physical condition, while the other was at the back with the rest of the participants. 

Distance travelled was not the main objective of the study; the goal was to foster social relationships with participants and enjoyment of outdoor activity. As an empowerment measure, each day a different participant led the excursion. To unite the group and discourage abandonment, a voluntary WhatsApp group was created for the outings to be remembered, for photos taken during the walks to be shared, and to periodically praise achievements and ask about those who did not come or were sick.

The control group received usual care at their primary care centre.

### 2.2. Ethics Approval and Consent to Participate

In accordance with Law 14/2007 of 3 July on biomedical research and human rights (BOE of 4 July 2007), all participants were informed and all signed a voluntary consent form to participate in the study. The study protocol was approved by the IDIAPJGol Clinical Research Ethics Committee (CEI 19/031-P).

### 2.3. Variables and Measurement Methods

1.Sociodemographic variables: sex, age, marital status (single, married, separated, widowed), living alone (yes/no), educational level (no education/primary education/secondary education/higher education). Basic health area: semi-urban (<15,000 inhabitants) or urban (≥15,000 inhabitants).2.Engage in regular physical activity (yes, no).3.Clinical remission of depression or response to intervention upon completion of the intervention. Clinical remission was defined as a Beck Inventory (BDI-II) score <14 points, and a response to the intervention was defined as a decrease from the baseline score [25].4.Remission of clinical anxiety or response to the intervention after the intervention is completed. Clinical remission was defined as a score <10 points on the GAD-7 scale (Generalised Anxiety Disorder), and a response to the intervention was defined as a reduction in the score from baseline [26].5.Improvements in social support once the intervention is over. A decrease in the DUKE-UNC-11 Social Support Questionnaire score from baseline, and good social support; a score of <32 points was considered an improvement in social support [27].6.An improvement in health-related quality of life (HRQoL) after the end of the intervention—a decrease in the score on the EuroQol questionnaire (EQ-5D) with respect to baseline was considered an improvement in HRQoL [28,29].7.Improved perception of health status after the intervention. An improvement in the perception of health status was considered an increase in the score on the scale.8.Outcome variables measuring viability in the intervention group:
-Satisfaction with the intervention: At the end of the intervention, a satisfaction survey was conducted, with 5 items and a 5-point Likert scale.-Adherence to the intervention: Attendance at walks was recorded. Adherence was calculated for the intervention variable (attendance at 75% or more of the sessions).-Number of visits made to the primary care centre: pre-intervention (4 months), from 2 November 2018 to 1 March 2019; intervention (4 months), 2 March 2019 to 1 July 2019; post-intervention (4 months), 2 July to 1 November 2019.9.Any variables that can act as confounders or effect modifiers:
-Pharmacological treatment. The DDD (defined daily dose, WHO) was calculated for each active ingredient, taking into account the number of days, the dose dispensed, and the route of administration of the drug. The active ingredients registered were those belonging to the antidepressant and anxiolytic groups.


### 2.4. Participant Timeline

Several different types of professionals participated in the fieldwork (Figure 1). The doctors or nurses to whom the patients were assigned acted as professional recruiters. Evaluators carried out baseline interviews at 4 and 12 months, but were blinded regarding patient group assignment. The researcher in charge of the random allocation of patients was not linked to any PCT. This researcher assessed the inclusion and exclusion criteria and performed the random allocation once all of the patients in the required sample consented to participate. There were two professionals for each PCT, who accompanied the group on their walks. The study also involved professionals from the sociocultural centres. 

An external evaluator was in charge of collecting the results of the previously mentioned paper-based self-administered questionnaires. The questionnaires were kept in a safe place, to which only the evaluator had access. The test evaluator read the questionnaires using an automatic reader and TeleForm Elite v 8.2 reading software (OpenText, Waterloo, ON, Canada).

### 2.5. Sample Size Calculation

The GRANMO calculator version 7.12 was used, https://www.imim.es/ofertadeserveis/software-public/granmo (accessed on 15 March 2017).

The sample size was determined for independent groups, able to detect a difference of 4.5 points in the mean score of the Beck Depression Inventory (deviation of 7 points [25]), 2.7 points in the mean score of the GAD-7 (deviation of 4.5 points [26]), and 7 points in the DUKE-UNC-11 social support questionnaire (deviation of 10.6 points [27]). A control was assigned for each case, assuming a statistical power of 80%, with a significance level of 5%. We anticipated a drop-out rate of 10%, and it was calculated that 49 patients per group were needed. 

### 2.6. Recruitment

The professionals participating in the study consulted the electronic medical record database of the patients on their list to obtain a list of patients who met the inclusion criteria (with an active diagnosis of depression (ICD-10-CM code: F32.0, F32.8, F32.9, F33.0) or anxiety (ICD-10-CM code: F41.2, F41.9) and little social support (ICD-10-CM code: Z60.2, Z60.4, Z60.9)). Subsequently, the doctor or nurse responsible for the patients reviewed the list and confirmed the selection of patients with active diagnoses, taking into account the inclusion and exclusion criteria. These patients were then invited to complete the questionnaires. Their responses were used to generate a final list of included patients, taking into account the cutoff points of the scales (i.e., BDI-II > 14, GAD > 10, or DUKE-UNC-11 > 32). 

### 2.7. Assignment of Interventions (for Controlled Trials)

Allocation: The researcher responsible for randomly assigning patients to the different groups was not associated with any of the PCTs. PASW Statistics for Windows, Version 18.0 (SPSS Inc., Chicago, IL, USA), was used to randomly allocate participants to the control and intervention groups. Patients with depression, anxiety, and little social support were equally assigned between the intervention and control groups. We also performed stratified randomisation according to age and sex to increase the generalisability of the results.

### 2.8. Data Analysis

For the description of qualitative and quantitative variables and percentages, the mean and standard deviation or the median, range, and 25th–75th percentiles were used.

Before the intervention, the baseline scores of the selected questionnaires were compared after randomisation of the patients with an independent Student’s *t*-test to verify that there were no significant differences. After the intervention, the mean scores of the different questionnaires in both groups were compared independently, using Student’s *t*-test for paired samples. A reduction in scores on the anxiety, depression, and social support scales was considered, and clinical remission of depression (Beck Inventory < 14), clinical remission of anxiety (GAD-7 < 10), and good social support (DUKE-UNC-11 < 32) were also considered.

To check the normality, this study used the skewness and kurtosis [30,31,32]. Absolute values of skewness greater than 3 and of kurtosis greater than 10 may indicate a problem with the normality, and values above 20 may indicate a more serious problem [33]. In 1995, West suggested that the absolute values of skewness and kurtosis should not be greater than 2 and 7, respectively. Based on this recommendation, the absolute values of the skewness and kurtosis of all the items in this study were within the acceptable ranges of <2 and <7, respectively.

Differences in the categorical variables were checked with the chi-squared test and Fisher’s exact test. Additionally, relative risk was estimated both crudely and adjusted for baseline values and sex (BDI-II, GAD-7, and social support).

Results showing differences with *p* < 0.05 were considered statistically significant. The intervals were all 95% confidence intervals. SPSS v.18 software (SPSS Inc., Chicago, IL, USA) was used for the statistical analysis.

## 3. Results

The recruitment of participants was carried out from 14 January 2019 to 15 February 2019. Ninety (90) participants met the inclusion criteria and were randomly divided into two groups: 45 in the intervention group (IG), who performed the physical activity programme consisting of walking two days per week as a group; and 45 in the control group (CG) (Figure 2). No significant differences were observed between the groups with respect to sociodemographic characteristics and the results of the health status, social support, anxiety, and depression questionnaires (Table 1). 

There were losses during the intervention period (from 1 March to 31 June)—9 in the control group and 11 in the intervention group. The causes of abandonment of the control group were the lack of interest in completing the final questionnaire in two participants, health problems in six participants, and being away from their habitual residence in one case. The reasons for leaving the intervention group were health problems in eight participants, having to care for dependent relatives in two cases, and not fitting into the intervention group in one case. Figure 2 shows the participant flow. 

Table 2 shows that both groups showed improved results on the anxiety and depression questionnaires, but the intervention group improved more, in the sense that their mean scores were below the diagnostic cutoff points for depression (Beck Scale < 14 points: no depression) and anxiety (GAD7 scale < 10: no anxiety), in contrast to the control group with scores above the cutoff points. In relation to social support, differences were observed between the groups, with the IG improving. The intervention group also had improved health perceptions after walking for two days per week for 4 months, with a final score of 73.73/100—an increase of 11.76 points from the initial score. All of these changes are visualised in Figure 3.

Figure 3 shows the values on the scales analysed at baseline and at the end of the study for the control and intervention groups.

Table 3 shows that the IG, after the intervention, was 59% more likely to have a remission of depression (Beck Scale score < 14 (no depression)) and 45% more likely to have a remission of anxiety (GAD7 Scale score < 10 (no anxiety disorder)) than the CG. 

The satisfaction level of the intervention group was very high, with a median of 25 and Q1–Q3: 21.25–25 (the maximum score of the questionnaire was 25). A short report on the intervention including the participants’ opinions was drawn up, https://drive.google.com/file/d/1p_LzZfOD7Pu9XoskRlhqlNEL1OK_nywQ/view (accessed on 2 October 2022). Adherence to the programme was high, with 75% of the participants attending 75% or more of the walks. In 90% of the cases, non-attendance was caused by health problems or surgical interventions. The WhatsApp group was very active during the intervention and remained active post-intervention, especially during the confinement of the COVID-19 pandemic.

In relation to the number of visits made to the primary care centre between the IG and the CG, three periods were compared: pre-intervention, intervention, and post-intervention. Table 4 shows that in the intervention group the number of visits decreased during the intervention and post-intervention periods compared to the pre-intervention period; however, the differences compared to the control group were not significant.

In relation to pharmacological treatment, comparing the prescriptions of antidepressants and anxiolytics pre-intervention (January 2019) and post-intervention (January 2020), it can be seen that in most patients there was no change in medication (*p*-value = 0.349). The only difference, although not significant, was that five participants in the CG required more medication compared to the IG (Table 5).

## 4. Discussion

In this study, we investigated the effects of a moderate-intensity physical activity programme in a group sample of patients over 64 years of age with anxiety, depression, and/or poor social support. The main findings were that at the end of the intervention there were differences between the two groups; the IG had improved scores in the social support, anxiety, and depression questionnaires, as well as improved perceptions of health. 

In relation to clinical anxiety and depression, both groups showed improved results, but the IG improved more, scoring below the diagnostic cutoff points for anxiety and depression and, therefore, experiencing remissions in clinical depression and anxiety. It should also be taken into account that the CG increased their consumption of drugs, which could act as a confounding factor. Our findings suggest that walking in a group two days per week for 4 months reduces clinical depression and anxiety by 59% (1.08 to 2.33) and 45% (1.02 to 2.07), respectively. These results are similar to the findings of a systematic review and meta-analysis of group walking interventions describing a mean difference reduction in depression scores with an effect size of −0.67(−0.97 to −0.38) [19]. We have not found studies using the same intervention (i.e., the same study population, duration, and type of physical activity) [22,34]. However, some have conducted similar physical activity interventions, such as the study by Patel et al., who examined the effects of a 3-month physical activity (walking) intervention on depressive symptomatology in underactive community-dwelling patients and found a positive association between increases in leisure-time physical activity and decreases in depressive symptomatology [35]. Von Berens et al. observed that a physical activity programme of 2–3 days per week for 6 months improved depressive symptoms in community-dwelling older adults with limited mobility [36]. Vancini et al. studied overweight and obese patients and observed that a moderate aerobic training programme of 1 h on three days per week for 8 weeks improved clinical anxiety, depression, and quality of life [37]. The study of De Oliveira, who studied the effects of physical activity on older people in the community, also showed improvements in quality of life, anxiety, and depression [38]. 

In relation to social support, in our study, differences were observed between the two groups after the intervention, improving in the IG compared to the CG. The WhatsApp group was very active during the intervention and remained active post-intervention—especially during the confinement of the COVID-19 pandemic, where some became sick, others were very lonely, and the participants gave one another a lot of support. Social networks helped to provide supportive relationships, bonding through the exchange of feelings, thoughts, and the perception of belonging to the group. A systematic review of 27 articles suggested that older people with greater social support for physical activity are more likely to engage in leisure-time physical activity. However, the high variability in the measurement methods used to assess both social support and physical activity in the included studies made it difficult to compare the studies [39]. 

In relation to perceptions of health, the IG improved after the physical activity programme, obtaining a final score of 73.73/100—higher than 70, which indicates a good state of health according to the specialised literature [40]. Gallé et al. also demonstrated that an exercise programme consisting of 1-hour sessions twice per week for a year improved health status perceptions in the elderly [41]. 

It is also suspected that the intervention reduces visits to the primary care centre, although the results are not conclusive—probably due to the small sample size. The participants rated the intervention very positively. As with other studies, adherence was high [19,22], and non-participation in the intervention was in most cases due to health problems. 

Our results may encourage the implementation of community interventions in this population group, who are more susceptible to social isolation and problems of anxiety and depression, for which medical treatments are not always effective. If we can demonstrate its effectiveness as a specific intervention, it could be incorporated into the social prescription services available at primary care centres. 

One of the main strengths of this study is that it was carried out under conditions that are akin to the usual clinical practice of primary care. As a result, it is an intervention that can be easily implemented in such an environment without the need for considerable organisational or structural modifications.

### Limitations

The main limitation of this study is related to recruitment difficulties due to the characteristics of the participants, i.e., elderly people with anxiety or depression and other frequent comorbidities that could prevent the initiation or completion of the study. This meant that the initially calculated sample size was not reached and, therefore, the power of the study was lower. Another limitation is that since only elderly patients without functional limitations and able to walk for one hour per day, two days per week were included—excluding those with worse functional status—the generalisability of the study is limited. Finally, the comorbidities presented by the patients in both groups were not taken into account.

## 5. Conclusions

The group physical activity programme improved clinical anxiety, depression, social support, group relevance, and health status perceptions in a sample of individuals over 64 years of age with anxiety, depression, and/or low social support. The level of satisfaction with the intervention was very high. Further studies are needed to confirm the preliminary results obtained in this limited sample. Future research will have to evaluate the long-term impacts of and adherence to this type of intervention.

## Figures and Tables

**Figure 1 healthcare-10-02203-f001:**
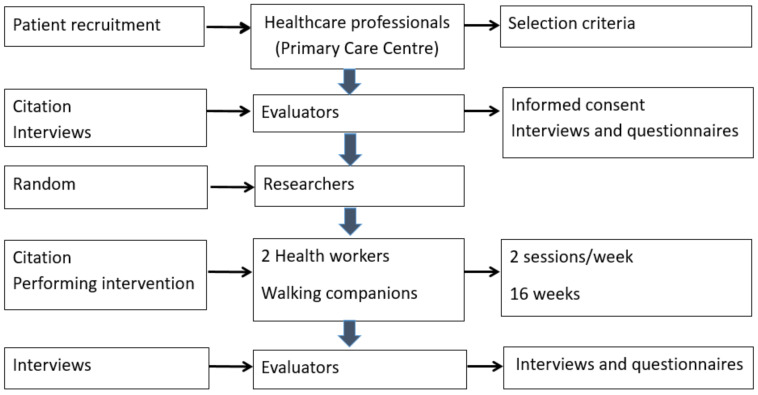
Flowchart of the intervention.

**Figure 2 healthcare-10-02203-f002:**
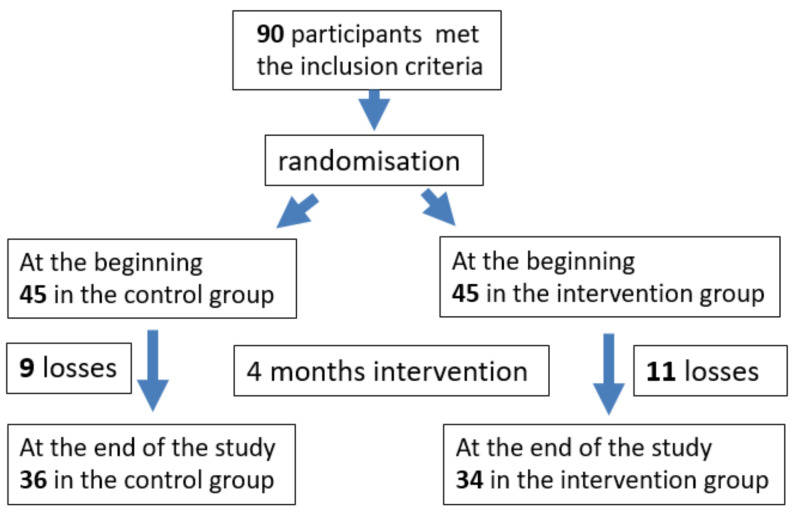
Participant flow.

**Figure 3 healthcare-10-02203-f003:**
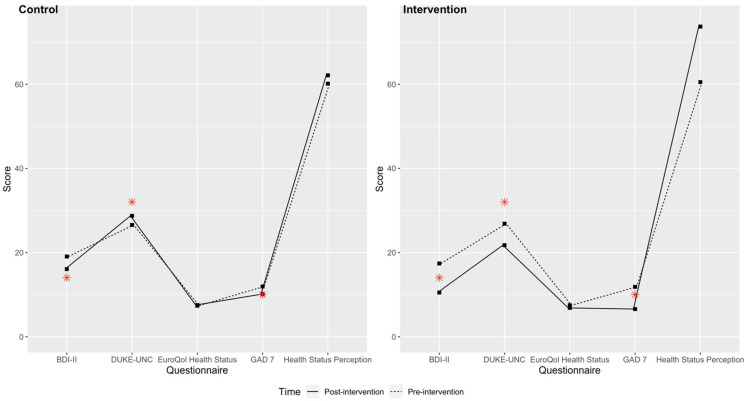
Values of the scales studied at baseline and at 4 months for the control and intervention groups. Depression (BDI-II), social support (DUKE-UNC), and generalised anxiety (GAD-7); asterisks in red indicate the cutoff points for the variables.

**Table 1 healthcare-10-02203-t001:** Baseline characteristics of the participants.

	*n* (%)		*p*-Value
	Control (*n* = 45)	Intervention (*n* = 45)	
Sex (% female)	81.81%	71.43%	0.376
Age (mean, SD)	74 (5.18)	75 (6.02)	0.632
Coexistence core			0.865
Lives with own family	28 (62.22)	28 (62.22)	
Lives with family of origin	1 (2.22)	2 (4.44)	
Lives alone	16 (35.56)	14 (31.11)	
Others	0 (0)	1 (2.22)	
Primary care team			0.833
Manresa (urban)	14 (31.11)	15 (33.33)	
Sant Joan de Vilatorrada (semi-urban)	15 (33.33)	16 (35.56)	
Súria (semi-urban)	16 (35.36)	13 (28.89)	
Educational level			0.905
No education	6 (13.33)	4 (8.89)	
Incomplete compulsory education	20 (44.44)	21 (46.67)	
Complete compulsory education	13 (28.89)	15 (33.33)	
High school or vocational training	5 (11.11)	3 (6.67)	
Completed university studies	1 (2.22)	1 (2.22)	
Others	0 (0)	1 (2.22)	
Marital status			0.322
Single	4 (8.89)	5 (11.11)	
Married or living with a partner	23 (51.11)	21 (46.67)	
Separated or divorced	3 (6.67)	4 (8.89)	
Widowed	15 (33.33)	11 (24.44)	
Others	0 (0)	4 (8.89)	
Engages in regular physical activity			1.000
Yes	17 (37.78)	17 (37.78)	
No	26 (57.78)	27 (60.00)	
Others	2 (4.44)	1 (2.22)	
Questionnaire results			
Health status—EuroQol (mean and SD) ^a^	7.38 (1.62)	7.47 (1.70)	0.798
Health status perception (mean and SD) ^a^	60.15 (18.11)	60.53 (21.18)	0.927
Social support—DUKE-UNC (mean and SD) ^a^	26.55 (11.05)	26.84 (12.29)	0.907
Social support—DUKE-UNC			0.823
Yes (<32)	31 (68.89)	29 (64.44)	
No (≥32)	14 (31.11)	16 (35.56)	
Generalised anxiety—GAD-7 (mean and SD) ^a^	11.93 (4.84)	11.84 (4.98)	0.931
Generalised anxiety—GAD-7			1
Yes (<10)	14 (31.11)	15 (33.33)	
No (≥10)	31 (68.89)	30 (66.67)	
Depression—BDI-II (mean and SD) ^a^	19.04 (7.21)	17.38 (7.44)	0.285
Depression—BDI-II			0.763
Yes (<14)	12 (26.67)	14 (31.82)	
No (≥14)	33 (73.33)	30 (68.18)	

(SD: standard deviation). Chi-squared or Fisher’s F contrast. ^a^ Student’s *t* contrast.

**Table 2 healthcare-10-02203-t002:** Post-intervention (4 months) questionnaire scores and changes in scores relative to baseline (pre-intervention) scores.

	Control (*n* = 36)					Intervention (*n* = 34)			Change Control vs. Intervention
	Value 4 m	Change	95% CI of Change	*p*-Value *	Value 4 m	Change	95% CI of Change	*p*-Value *	*p*-Value
EuroQol Health status	7.53 (1.56)	0.08 (1.23)	(−0.33; 0.49)	0.686	6.85 (1.54)	−0.38 (1.54)	(−0.92; 0.15)	0.156	0.168
Health status perception	62.14 (14.98)	1.83 (16.92)	(−0.39; 7.56)	0.520	73.73 (18.51)	11.76 (18.64)	(5.26; 18.27)	<0.001	0.023
DUKE-UNC	28.72 (11.93)	2.97 (9.81)	(−0.35; 6.29)	0.078	21.73 (10.38)	−3.59 (11.68)	(−7.66; 0.49)	0.082	0.013
GAD-7	10.08 (4.69)	−1.83 (4.48)	(−3.35; −032)	0.019	6.59 (4.21)	−5.11 (5.73)	(−7.12; −3.12)	<0.001	0.009
BDI-II	16.05 (8.21)	−2.86 (5.64)	(−4.77; −0.95)	0.004	10.50 (6.40)	−6.36 (8.27)	(−9.29; −3.43)	<0.001	0.046

* Paired *t*-test. The change is the final value—initial value and is shown as the mean and standard deviation.

**Table 3 healthcare-10-02203-t003:** Results of the post-intervention questionnaires according to the cutoff points and relative risk (RR); *n* = 35 control group and *n* = 34 intervention group.

Depression (BDI-II)	≥14	<14	RR Crude	RR Baseline Adjusted
- Control	19 (52.8)	17 (47.2)	-	-
- Intervention	8 (23.5)	26 (76.5)	1.62 (1.09; 2.39)	1.59 (1.08; 2.33)
**Anxiety disorder (GAD-7)**	≥10	<10	RR	RR
- Control	18 (50.0)	18 (50.0)	-	-
- Intervention	8 (23.5)	26 (76.5)	1.53 (1.05; 2.23)	1.45 (1.02; 2.07)
**Social support**	≥32	<32	RR	RR
- Control	16 (44.4)	20 (55.5)	-	-
- Intervention	8 (23.5)	26 (76.5)	1.37 (0.97; 1.95)	1.36 (0.99; 1.88)

**Table 4 healthcare-10-02203-t004:** Comparison of visits to the CAP between the GG and the GI.

Group	Period Pre-Intervention 1 November 2018 to 28 February 2019 N = 45 Both Groups		Intervention Period 1 March 2019 to 30 June 2019 N Control = 36, n Intervention = 34		Period Post-Intervention 1 July 2019 to 31 October 2019 N Control = 36, n Intervention = 34	
	Total Count	Mean (SD)	Total Count	Mean (SD)	Total Count	Mean (SD)
Intervention	258	7.59 (5.78)	223	6.76 (5.74)	224	7.23 (8.15)
Control	205	5.86 (3.67)	248	7.09 (5.01)	213	6.26 (4.34)
Comparison *	0.137		0.798		0.533	

* Student’s *t*-test to compare the intervention group and the control group.

**Table 5 healthcare-10-02203-t005:** Comparison of pharmacological treatments between the GG and GI—absolute frequencies and percentages.

Control (*n* = 36)			Intervention (*n* = 34)		
No Change	More Medication	Less Medication	No Change	More Medication	Less Medication
25 (69.44)	8 (22.22)	3 (8.33)	28 (82.35)	3 (8.82)	3 (8.82)

## Data Availability

The data that support the findings of this study are available from the corresponding author upon request.

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
