# Peer review of "Effects of Physical Activity Interventions in the Elderly with Anxiety, Depression, and Low Social Support: A Clinical Multicentre Randomised Trial"

_healthcare, 2022, doi:10.3390/healthcare10112203_

Round 1

Reviewer 1 Report

Major comment:

The authors should use standardized guidance on what should be included in reporting a randomized controlled trial. For example, the CONSORT statement (http://www.consort-statement.org/) that designed for reporting randomized clinical trials. Based on the CONSORT statement, this paper is missing some important components and needs to be extensively revised before reconsideration for publication. For example, the first item on the CONSORT checklist is the title.

Title: To help ensure that a study is appropriately indexed and easily identified, authors should use the word "randomized" in the title to indicate that the participants were randomly assigned to their comparison groups.

Minor comments: 

The authors may work more on improving the writing of this manuscript. Some examples as below.

“After the intervention there were positive differences in the intervention group (IG), with improvement in depression, anxiety, health status perception 29 and social support.”

Should be either “positive differences between the groups..” or “significant improvements in the IG”?

“The possibility of a remission of depression and anxiety was 62% and 53% respectively.”

Should be “62% and 53% higher…”?

Author Response

Attached response to the reviewer

Reviewer 2 Report

Dear authors, I consider that the work has difficulties for publication with essential quality deficiencies. The sample is scarce for a study of these characteristics, the procedure is missing, the methodology does not require a class of methods but the specification of which test has been used and why, the levene or homogeneity, kurtosis, distribution, etc.

 The results need more analysis, as they do not offer much progress, largely due to the small sample.

The discussion and conclusions should be restructured once the analyses are more advanced, for the quality of this journal.

Author Response

Attached response to the reviewer 2

Reviewer 3 Report

Dear Authors

After reviewing your manuscript, I would like to raise some questions:

1.- Methodology: when calculating the sample size per group, could you please indicate the software used? Also, could you please indicate why you have not included the replacement rate for possible losses in the sample?

2.- Analysis- It should indicate the use of tests that validate the normality of the sample. I could not find that information in your article, please Could you please include this information?

Could you explain the reason for using parametric tests to compare measures from tests that provide numerical values but are representations of something qualitative?

3.- Ethical aspects of the study. Authors should include not only the information that the study has been evaluated and assessed by an ethics committee but also that all participants were informed and all signed a voluntary consent to participate in the study.

Please include a section on methodology with this information.

Author Response

Attached response to the reviewer

Round 2

Reviewer 1 Report

The authors have adequately addressed most of the concerns. However, there are a few editorial issues remain. The authors should have some professionals review the manuscript and correct all the grammar mistakes. Some examples are below:

  1. The title should be “Effect of physical activity intervention in elderly with anxiety, depression and low social support: a clinical multicentre randomized trial”      

  2. The use of and/or is severely frowned upon in formal writing. Consider using only one conjunction or rewriting.

  3. It should be “The questionnaires were kept in a safe place to which only the evaluator has access.” instead of “The questionnaires was…”

  4. “During the intervention period (from March 1 to June 31), there were losses, 9 in the control group and 11 in the intervention group.” This sentence may be hard to follow. Consider “There were losses during the intervention period (from March 1 to June 31), 9 in the control group and 11 in the intervention group.”

  5. “The reasons for leaving the intervention group were health problems in 8 participants, having to to care for dependent relatives in two cases and not fitting into the intervention group in one case.” Please remove the repeated “to”.

Author Response

We attach the response to the reviewer

Reviewer 2 Report

It still has not adequately resolved the restructuring of the discussion and has not responded to that point of the review.

Author Response

We attach the response to the reviewer
